# Changes in Salivary Levels of Creatine Kinase, Lactate Dehydrogenase, and Aspartate Aminotransferase after Playing Rugby Sevens: The Influence of Gender

**DOI:** 10.3390/ijerph17218165

**Published:** 2020-11-05

**Authors:** Álvaro González Fernández, Jose Enrique de la Rubia Ortí, Lorena Franco-Martinez, Jose Joaquín Ceron, Gonzalo Mariscal, Carlos Barrios

**Affiliations:** 1Institute for Research on Musculoskeletal Disorders, Catholic University of Valencia San Vicente Mártir, 46001 Valencia, Spain; alvaro.glez@mail.ucv.es (Á.G.F.); carlos.barrios@ucv.es (C.B.); 2Department of Nursing, Catholic University of Valencia San Vicente Mártir, 46001 Valencia, Spain; joseenrique.delarubi@ucv.es; 3Interdisciplinary Laboratory of Clinical Analysis, Campus of Excellence Mare Nostrum, University of Murcia, 30100 Murcia, Spain; lorena.franco2@um.es (L.F.-M.); jjceron@um.es (J.J.C.)

**Keywords:** rugby sevens, muscle damage, gender, creatine kinase, lactate dehydrogenase, aspartate aminotransferase

## Abstract

Rugby sevens is characterised by continuous exertion and great physical contact per unit of time, leading to muscle damage. It is important to identify markers that can quantify muscle damage in order to improve recovery strategies. The objective of this study was to evaluate the release dynamics of muscle damage markers creatine kinase (CK), lactate dehydrogenase (LDH), and aspartate aminotransferase (AST) in saliva samples when playing rugby sevens, analysing the influence of gender, during the rugby sevens university championship of Spain. The total sample included 27 athletes, divided into two teams of 14 men and 13 women between 18 and 31 years of age. CK, LDH, and AST were quantified from salivary samples collected from each athlete before and after three rugby sevens matches. The modified Borg scale of perceived exertion was also used after each match. When the results were analysed globally, there were no differences in CK and LDH before and after any match, but AST did show differences after two days of completing all matches. In terms of gender, the three enzymes showed different responses in men and women. Regarding the Borg scale, there were only significant differences between men and women after completing all mataches, with a greater perceived exertion in women. Based on our results, it can be stated that that serial matches of rugby sevens can cause changes of different magnitude in AST, CK and LDH activities in saliva, with AST showing the most significant variations and these changes are more pronounced in men than in women.

## 1. Introduction

Rugby is currently one of the most played and followed contact sports in the world [1]. There are two main kinds of rugby: rugby union and rugby sevens. In particular, rugby union is a team sport with great physical demand, including high intensity activities, such as sprinting, rucks, scrums, and tackles. It also includes low intensity activities, such as walking and jogging [2]. Rugby sevens is a traditional sport, but more attention has been paid to this form of rugby since it was included in the Olympics. In this modality, there are seven players on each team (three forwards and four backs), and the game lasts 15 min in total (two halves of seven minutes and a one-minute rest at the half-time break). This version, regarding Rugby League, increases both aerobic and anaerobic activity even more [3] with high speed fatigue [4], especially more than observed in rugby union. However, despite players possibly suffering less contact cruelty in narrow spaces than in rugby union, the fact that many rugby sevens competitions involve several matches played on consecutive days leads to greater impact in terms of neuromuscular damage and recovery [5]. Efficient muscle damage markers need to be identified in order to propose suitable recovery strategies due to this great contact. In addition, regarding these recovery strategies, rugby sevens has more and more followers, with exponential growth in the number of competitions and athletes who take part in both male and female elite competitions [6]. As a result, possible differences in muscle damage depending on gender would also be interesting. Therefore, recovery strategies can be adapted and, in this sense, athletes’ performance would be improved.

Aetiology of muscle damage caused by physical exercise is mainly based on three mechanisms: mechanical fibre interruption, alterations of calcium homoeostasis, and inflammatory processes [7]. Within the inflammatory processes, measuring certain biomarkers in serum and plasma has been used to identify muscle damage. Among these biomarkers, the enzymes creatine kinase (CK), lactate dehydrogenase (LDH), or aspartate aminotransferase (AST) stand out as tools to determine skeletal muscle injury and tissue damage in the muscles [8]. These enzymes are found in the cytoplasmic matrix of muscle cells; therefore, their presence in serum or plasma are an indicator for cell lesions [9]. CK is considered to be one of the most important markers for muscle damage [10]. High levels of LDH specifically indicate muscle fatigue, while high levels of AST in the blood can be due to a wide range of clinical alterations; therefore, it is regarded as a more non-specific biomarker [11]. Quantification of these three molecules (CK, LDH, and AST) as markers of the response to exercise has not only been performed on blood samples but also from salivary samples [12]. In this sense, quantification in saliva has advantages over blood samples, highlighting that sample collection is a simple, safe, inexpensive, and non-invasive technique, thus reducing anxiety and discomfort in comparison to extracting blood [13]. From an analytical point of view, saliva contains less proteins than serum, so there is less risk of non-specific interference [14].

Regarding muscle damage response to physiological stress according to gender, we know that women are more resistant to neuromuscular fatigue during isolated isometric muscle contractions and of a similar intensity in comparison to men [15]. Nonetheless, if these contractions were caused with dynamic tasks, the answer is less clear, as they depend on the specific task and, therefore, on variables, such as shortening velocity and muscle group. In addition, rugby sevens players can be affected more uniformly by this contact cruelty and fatigue than in rugby union. It is for this reason that further studies are necessary to research the response in muscle damage according to the activity. In particular, this article delves into this point. The approach and originality of the study is to describe the kinetics of muscle damage by quantifying some related biomarkers after high intensity physical activity over two days, as represented with rugby sevens. Thus, the aim of this study is to establish the release dynamics of the muscle damage markers of the enzymes CK, LDH, and AST in saliva samples when playing rugby sevens, additionally to establishing the possible differences according to gender, which could establish different recovery strategies.

## 2. Material and Methods

### 2.1. Study Design

This is a cohort, observational, longitudinal, quantitative, and prospective pilot study with follow-up.

### 2.2. Subjects

The samples have been obtained from all the university rugby sevens players at the Catholic University of Valencia who voluntarily agreed to take part in this study. The study was carried out during the final stages of the rugby sevens university championship of Spain, both for women’s and men’s teams. All participants signed an informed consent form after being informed on the nature of the study. The total sample included 27 individuals between 18 and 31 years of age. Two cohorts were created: a male cohort including 14 athletes that were on the men’s rugby sevens university team, and the female cohort including 13 athletes that were on the women’s rugby sevens university team. All participants in the study followed the exact same guidelines for recovery and diet throughout the competition (without differentiation between men and women). These guidelines were designed and controlled by nutritionists and sports recovery specialists, both specialised in top-level sports competitions.

### 2.3. Procedure

The competition took place in Valencia in May 2019, with an environmental temperature of 24 °C. Two different championships were held, a women’s and a men’s. Six samples of saliva were collected from each participant during the tournament. As seen in Figure 1, the first sample was collected before match 1 on day 1 of the competition (9 a.m.) and the second aftermatch 1 (10 a.m.). The third and fourth samples were taken before and after match 2 (4:20 p.m. and 5:20 p.m., respectively). Finally, the fifth and sixth samples were obtained the following day, before and after each team’s match 3 (11:00 a.m. and 12:00 p.m., respectively). The samples of saliva were collected 5 min before and 5 min after each match. Samples were collected from all participants in the study who needed to rinse their mouths with distilled water in order to avoid altering the samples with food left in the mouth that could contain a high content of acid or sugar. At least 2 mL of unstimulated saliva was collected in 10 mL sterile plastic tubes that were then stored in a container with ice. The procedure had an approximate total duration of 5 min. The samples were then centrifuged in the laboratory at 1500 g for 15 min, and the supernatant was frozen and stored in micro sample tubes at −20 °C until the samples were ready to be analysed.

In order to quantify the intensity of the exercise, perceived intensity was measured 10 min after exertion of each match using the modified 0–10 Borg scale, where 0 represents no exertion and 10 very, very hard exertion [16]. The players were familiar with this method before data were collected.

The players’ position was not taken into account in this study due to the fact that physiological and performance characteristics in rugby sevens are relatively homogeneous in the different positions of the game [3,17], which is possibly associated with movement patterns in rugby sevens being less position-dependent than in rugby union [18].

Nonetheless, the 27 players took part in all the matches for at least 7 min and for a maximum of 9 min each.

### 2.4. Analysis of Salivary Samples

Immediately after collection, the samples were centrifuged (500× *g* for 10 min at 4 °C) and stored in a freezer at −80 °C until needed for analysing. Once the samples were thawed, they were thoroughly vortexed before analysis. CK was quantified by means of a commercial kit (Beckman Coulter, Brea, CA, USA), as well as LDH with a commercial kit (Biosystems, Barcelona, Spain) and AST with a commercial kit (Beckman Coulter, Brea, CA, USA). All tests were adapted in order to be used to collect saliva samples and were carried out in an automated biochemical analyser (Olympus A400, Beckman Coulter, Brea, CA, USA) at 37 °C.

### 2.5. Statistical Analysis

The statistical analysis was carried out with the SPSS 25 package for Windows (SPSS Inc., IBM Company, Armonk, NY, USA), with a level of significance of *p <* 0.05. After establishing the non-normality of the sample with the Kolmogorov-Smirnov test, the nonparametric Mann–Whitney U test was used and applied to two independent samples, while the Wilcoxon signed-rank test was used for related samples.

### 2.6. Ethical Concerns

The study was developed in accordance with the Declaration of Helsinki [19] with the prior approval of the protocol by the Ethical Committee of the Catholic University of Valencia (UCV/2019-2020/017). Participants were provided with a written informed consent form after being informed on the procedures and the nature of the study.

## 3. Results

The sociodemographic characteristics of the population sample of the study are demonstrated in Table 1, where no significant differences between men and women in terms of age were observed, yet, in terms of weight, a differentiation can be seen (Z = −3.88; *p <* 0.05), height (Z = −3.88; *p <* 0.05), and body mass index (BMI) (Z = −2.18; *p* = 0.029).

When saliva enzymes were analysed in the whole sample of 27 athletes, CK and LDH did not show significant differences in activity before and after any of the 3 matches (Table 2 and Table 3). However, there was a significant increase in AST after match 3 on the second day (Table 4).

Regarding gender, when comparing values between the men’s and women’s teams before and after each match, we could observe that CK showed significantly higher values in men before match 2 and after the match 3 (Table 2). AST, although showed significant increases after match 3 form men and women, it has significantly higher values in the men’s team that in women after playing the three matches. (Table 4). LDH did not show significant differences between men and women in any of the three matches (Table 3).

As far as the Borg scale, the mean values throughout the competition for men and women varied between 5.42 ± 1.34 and 7.25 ± 1.48, illustrating a perceived exertion between hard and very hard, respectively. Concerning differences between the men and women, there were only significant differences (*p <* 0.008) in match 3 with a higher perceived exertion in women (Figure 2).

## 4. Discussion

Rugby sevens is characterised by quick and powerful muscle contractions, as well as continuous collisions between players, that can lead to muscle damage. In this sense, measuring activity of CK, LDH, and AST in serum can be used as a muscle injury marker [9], as they have been observed to increase in the blood after physical activities that cause intense muscle fatigue, such as running a mountain marathon [20] or resistance exercises [21]. These enzymes can also be quantified in saliva [22]. Therefore, CK, LDH, and AST were quantified in our study in saliva samples obtained before and after the rugby sevens matches. Despite the lack of studies of this nature that identify normal values of enzymes in saliva related to possible muscle damage, the values obtained in our study for AST and CK were higher than those described by Barranco T. et al., 2018, in healthy people and who had not performed any kind of previous exercise [23].

When analysing the possible changes in these enzymes total study population, no significant differences were observed for CK and LDH concentration levels before and after any of the 3 matches, despite the results of the Borg scale indicating a perceived exertion between hard and very hard, respectively. These results do not coincide with those obtained in another study, where there were significant changes in saliva after playing an indoor football match, registering an equally high perceived exertion (average of 7 on the Borg Scale) [24]. The fact that these results do not coincide could be due to the time when the samples were collected, as samples were taken immediately after physical activity in our study. However, in the study conducted by Barranco et al. [12], the samples were collected 30 min and 12 h after exertion. In this sense, we must outline that there has been evidence of a delay in the movement of some enzymes from blood to saliva, including CK [25]. In addition to this, it is precisely CK that takes time to increase in the blood after physical activity, as other authors observed a rise in plasma after 72 h after a football match [26]. In particular, when CK was measured after playing rugby, it was only increased in the blood after 30 min after the match [27], remaining high and even showing higher peaks after 24 h after finishing the sport [27,28]. Something similar was also registered for LDH, whose levels were seen to increase in the blood after 8, 24, and 48 h after exercise, coinciding with the feeling of muscle ache [29]. Yet, in saliva, the only published data indicate an increase 4 weeks after the aerobic exercise was carried out [30]. Nonetheless, we did observe a significant increase in AST after match 3 on the second day in our study. Curiously, this enzyme was seen to rise in the blood samples obtained immediately after a high intensity boxing match [31], which seems to confirm our results.

Nonetheless, this study has also assessed the possible differences in the response of the 3 enzymes according to gender. Authors, such as Franco Martínez L. et al. [32] and Souglis A. et al. [33,34], show in their studies that there is a different salivary proteome in men and women after exercising. They propose the idea to conduct further studies to learn what physiological changes are represented in the saliva of both men and women. In this sense, we observed that CK, AST, and LDH behaviour is different throughout the rugby sevens competition, showing generally higher enzyme levels in men, both before and after the matches.In particular, if the differences between pre- and post-match values between both men and women are analysed, CK shows higher levels in men before match 2 and after match 3. Something similar happens with AST, as match 1 displays these significant differences are always higher after the match. LDH is the only case where there are no differences in any match. This would indicate a different behavior of the enzymes analysed in our study according to gender. 

Despite not being able to compare these results with previous ones, due to the lack of research analysing differences between men and women in terms of enzyme salivary secretion after physical activity, they were in line with a recent study where the values obtained in serum samples were compared. Specifically, after the Ecomotion Pro edition, significant increases in CK, AST, and LDH in serum were detected only in men, showing differences between men and women in the secretion of these enzymes [35]. These results show that behaviour in the face of stress is different in men and women. In addition the study conducted by O’Leary T.J. et al. [36] differences in the response to physiological stress afterload carriage was dectected regarding gender. It was specifically observed that, despite the fact that women had a higher heart rate and greater perceived exertion after activity, loss in strength of maximal isometric contraction was higher in men, therefore higher physiological stress during load carriage for women did not mean more severe muscle fatigue, showing more resistance towards said fatigue [36] and confirming what had been described in another prior study [15].

With regard to the Borg scale, perceived exertion implies a subjective assessment of strength, tension, discomfort and/or fatigue while exercising. Therefore, perception depends on physiological mediators grouped into central mediators (related to cardiorespiratory processes) and peripheral mediators (related to processes inherent to skeletal muscle, with blood acidosis, etc.) [37]. Concerning central mediators, important correlations between perceived exertion and heart rate have been established (between r = 0.80 and r = 0.90) [38]. However, this correlation is not as obvious in peripheral mediators, such as biomarker quantification. Thus, a close relation between concentration of blood lactate and the valuation of perceived exertion are established [39]. Nonetheless, other authors have found differences up to 3–4 points on the perceived exertion scale for the same value of concentration of blood lactate (4 mmol/L), in a maximal, progressive treadmill test [40]. This discrepancy between biomarkers and perceived exertion would be in line with our results, where female players show a higher perceived exertion than men, even though the levels of muscle damage markers are generally higher in men. On the other hand, delving into discrepancies in perceived exertion between men and women regardless of muscle damage, it does seem that it is higher in women, in accordance with studies published in other situations involving intense physical activity [41]. These results seem to be in accordance with those obtained in our study, where women show a significantly higher perceived effort than men in match 3.

The results of our study showed a high intersubject variability. In this sense, other authors have described that the values of muscle enzymes show this great variability between individuals and depend on the individual to a high degree, observing high variability between athletes who are even on the same team. This fact demonstrates the need for individualised follow-ups [42,43]. In terms of rugby, this variability has been previously verified in other enzymes, such as myoglobin. Lindsay A. et al. suggest that each subject should be analysed individually and not as a group, as rugby is a sport where muscle damage is highly variable, depending on the progression of the match [44]. Moreover, although this study used two complete teams, it would be necessary to increase the sample for future research as it is limited, analysing also the possible influence of match time on athletes. In addition since muscle damage leads to increase pain, this variable could be assessed through tests, such as vertical jumps, maximum isometric force, and subjective muscle pain measurements. Nonetheless, it would be interesting to do a longer follow up of the enzymes of our study after match in order to evaluate long-term changes in these analytes. 

In terms of the interest that our study may have, the differences in the response to physiological stress may be related to the onset of injuries after exercise. In skeletal muscle, there have been described gender differences in muscle injuries with a decrease in the frequency of injuries in female rodents [45], with CK having lower increases in serum after injury in women than men [46] that would be in agreement with our findings. The enzymes CK, LDH (to a lesser degree as it seems less sensitive), and especially AST could be used as possible diagnostic tools to predict this risk of injury, which could help to improve young athletes’ performance. 

## 5. Conclusions

CK, AST, and LDH in saliva showed changes of different magnitude after serial matches of rugby seven, with AST showing the most significant variations. Differences in terms of gender were observed, with men showing increases of higher magnitude, mainly in AST after the matches. Due to the large variations in enzyme values between athletes, it would be recommended to monitor the changes so they can be carried out individually in order to determine the status and muscle damage more objectively.

## Figures and Tables

**Figure 1 ijerph-17-08165-f001:**
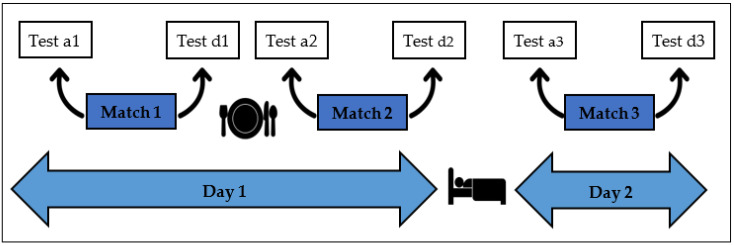
Timeline followed throughout both competitions (men and women) when collecting saliva samples.

**Figure 2 ijerph-17-08165-f002:**
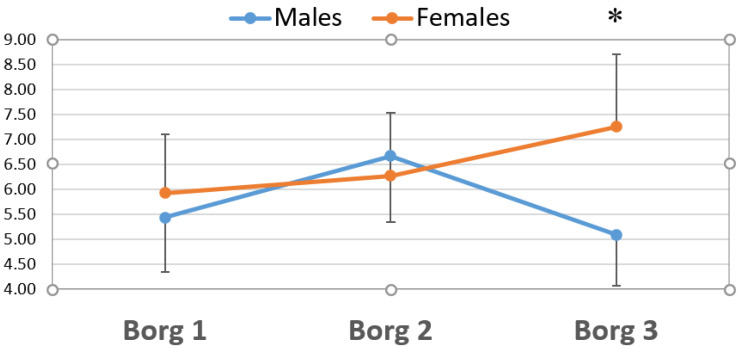
Comparison between men and men of perceived exertion as established by the modified Borg scale after the 3 matches. Borg: modified Borg scale of perceived exertion; * significant differences *p* < 0.05 with the Mann–Whitney U Test.

**Table 1 ijerph-17-08165-t001:** Anthropometric and age data of the study population.

	Men’S Team (*n* = 14)	Women’S Team (*n* = 13)	Mann–Whitney U Test	
	Mean ± SD	Mean ± SD	*Z*	*p*
Age (years)	22.21 ± 3.07	21.85 ± 2.27	−0.25	0.807
Weight (kg)	81.04 ± 8.51	64.93 ± 13.17	−3.49	0.000 ***
Height (cm)	175.56 ± 7.32	162.38 ± 4.72	−3.88	0.000 ***
BMI (Kg/m^2^)	26.27 ± 1.94	24.51 ± 3.81	−2.18	0.029 *

BMI: body mass index; * significant differences *p <* 0.05; *** significant differences *p <* 0.001.

**Table 2 ijerph-17-08165-t002:** Statistical analysis of creatine kinase (CK) samples in saliva.

CK (ng/mL)	Total Sample Mean ± SD (Median)	Men Mean ± SD (Median)	Women Mean ± SD (Median)	Mann–Whitney U Test Z (*p*)
1A	68.815 ± 51.853 (73.5)	84.977 ± 62.088 (85.5)	52.654 ± 34.360 (45.3)	−1.436 (0.151)
1D	88.512 ± 82.388 (47.0)	108.136 ± 106.300 (68.4)	65.617 ± 31.787 (34.9)	−1.157 (0.247)
Wilcoxon signed-rank test Z (*p*)	−1.457 (0.145)	−0.524 (0.600)	−1.778 (0.075)	
2A	61.005 ± 51.114 (42.9)	92.489 ± 63.105 (86.1)	35.246 ± 12.493 (30.1)	−2.545 (0.011 *)
2D	74.515 ± 59.811 (58.1)	106.656 ± 76.478 (75.4)	48.218 ± 20.675 (53.9)	−1.557 (0.119)
Wilcoxon signed-rank test Z (*p*)	−1.923 (0.055)	−0.889 (0.374)	−1.867 (0.062)	
3A	59.744 ± 55.243 (58.9)	77.115 ± 69.830 (63.0)	40.925 ± 24.411 (36.2)	−1.034 (0.301)
3D	78.280 ± 69.106 (47.0)	109.069 ± 83.919 (73.3)	44.925 ± 20.071 (40.2)	−2.448 (0.014 *)
Wilcoxon signed-rank test Z (*p*)	−1.789 (0.074)	−1.852 (0.064)	−0.706 (0.480)	

A: before the match; D: after the match; ***** significant difference *p <* 0.05.

**Table 3 ijerph-17-08165-t003:** Statistical analysis of lactate dehydrogenase (LDH) samples in saliva.

LDH (U/l)	Total Sample Mean ± SD (Median)	Men Mean ± SD (Median)	Women Mean ± SD (Median)	Mann–Whitney U Test Z (*p*)
1A	320.358 ± 299.804 (185.2)	341.000 ± 309.482 (209.3)	299.715 ± 300.919 (146.2)	−0.590 (0.555)
1D	360.635 ± 436.736 (192.2)	398.564 ± 538.385 (158.2)	316.383 ± 294.907 (106.2)	−0.669 (0.504)
Wilcoxon signed-rank test Z (*p*)	−0.175 (0.861)	−0.035 (0.972)	−0.078 (0.937)	
2A	306.565 ± 328.447 (201.2)	423.389 ± 423.862 (451.8)	210.982 ± 197.249 (132.7)	−0.570 (0.569)
2D	261.165 ± 252.243 (290.3)	377.089 ± 236.488 (328.5)	199.046 ± 161.811 (163.8)	−0.722 (0.470)
Wilcoxon signed-rank test Z (*p*)	−1.400 (0.161)	−1.481 (0.139)	−0.622 (0.534)	
3A	171.076 ± 250.427 (188.4)	134.792 ± 240.492 (70.7)	210.383 ± 265.507 (84.3)	−0.870 (0.384)
3D	279.904 ± 372.832 (220.1)	797.008 ± 488.854 (223.3)	153.042 ± 93.875 (122.5)	−0.598 (0.550)
Wilcoxon signed-rank test Z (*p*)	−1.816 (0.069)	−2.691 (0.007 ^+^)	−0.392 (0.695)	

A: before the match; D: after the match; **^+^** significant differences *p <* 0.05.

**Table 4 ijerph-17-08165-t004:** Statistical analysis of aspartate aminotransferase (AST) samples in saliva.

AST (U/l)	Total Sample Mean ± SD (Median)	Men Mean ± SD (Median)	Women Mean ± SD (Median)	Mann–Whitney U Test Z (*p*)
1A	45.992 ± 57.848 (30.6)	67.700 ± 74.819 (36.8)	24.285 ± 18.781 (25.8)	−2.333 (0.020)
1D	57.727 ± 80.259 (19.1)	81.686 ± 101.581 (37.3)	29.775 ± 19.424 (15.2)	−1.260 (0.208 *)
Wilcoxon signed-rank test Z (*p*)	−0.646 (0.518)	−0.210 (0.834)	−0.941 (0.347)	
2A	42.530 ± 58.500 (18.3)	73.389 ± 78.034 (45.4)	17.282 ± 8.521 (14.6)	−1.824 (0.068)
2D	57.325 ± 92.423 (26.8)	99.300 ± 128.574 (45.6)	22.982 ± 11.026 (20.5)	−2.013 (0.044 *)
Wilcoxon signed-rank test Z (*p*)	−1.493 (0.135)	−0.889 (0.374)	−1.600 (0.110)	
3A	27.896 ± 37.218 (16.0)	41.262 ± 47.301 (43.1)	13.417 ± 11.900 (15.4)	−1.251 (0.211)
3D	47.496 ± 53.572 (22.1)	71.785 ± 65.588 (39.4)	21.183 ± 10.791 (18.4)	−2.284 (0.022 *)
Wilcoxon signed-rank test Z (*p*)	−2.879 (0.004 ^++^)	−2.201 (0.028 ^+^)	−1.962 (0.048 ^+^)	

A: before the match; D: after the match; **^+^** significant differences *p <* 0.05, **^++^**
*p <* 0.005; ***** significant differences *p <* 0.05.

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
