# Peer review of "Changes in Salivary Levels of Creatine Kinase, Lactate Dehydrogenase, and Aspartate Aminotransferase after Playing Rugby Sevens: The Influence of Gender"

_ijerph, 2020, doi:10.3390/ijerph17218165_

Round 1
Reviewer 1 Report
Dear authors,
I have read the manuscript with interest about candidate saliva biomarkers of muscle damage related to intense physical exercise (rugby players). The goal of the study is relevant and can shed light on the relation of muscle damage with various types of sports. The ‘Introduction’ section of the manuscript is well written and puts this study ‘in-context’.
1-My main worry about this study is the high variability of data. I definitely understand and expect a variation between samples. However, reaching conclusions is very difficult by using highly variable data. Do the authors have an explanation on the data variability?
2-Although there is high standard deviation, all of the values are lower in 1D compared to 1A. Any explanation on this? why is this small drop?
3-The design of the study is valid and having 6 time points is acceptable. However, one wonders, if it is possible to include a new sample population and following the subjects for longer? Second paragraph of the ‘Discussion’ includes some information related to that. However, additional data can really support this manuscript by following the subjects for longer period of time. Do the authors consider studying another population of rugby players and following for longer hours after the match?
4-Apart from these, the authors made several conclusions on the increase of several measurements after the 3rd match. For example, the increase in AST (41 to 71 in men, 13 to 21 in women). I am not convinced that these increases are physiologically meaningful since these values are still lover that the initial measurements in 1A (71 is less than 81 in men, 21 is less than 29 in women). Any explanation on these observations?
5-Due to the above points listed, I invite the authors to tune-down some of the conclusion.
6-Although the discussion session has a lot of useful information, I expected to find comparison of these values with healthy subjects. For, example what are the normal CK, LDH and AST values in the saliva of healthy individuals? Are there studies reporting these? Or do authors consider performing these analysis? Additionally, are there studies looking at these parameters in patients with muscle diseases? If so, what are these values. I think, both of these can show if the observed differences are biologically meaningful or not.
7-A small point on the formatting of the study scheme. Please, make sure that all is in-line and readable (e.g. Match1, Match2, Match3).And the figure legend is missing.
Author Response
Reviewer 1
1-My main worry about this study is the high variability of data. I definitely understand and expect a variation between samples. However, reaching conclusions is very difficult by using highly variable data. Do the authors have an explanation on the data variability?.
We understand the editor’s concern. However, other authors have already described that values of muscle enzymes have a high variability between individuals and dependon each athlete to a high degree, showing great variability between athletes even on the same team. We have added this information to the discussion section, indicating that this is a limitation for the use of enzymes in muscle status monitoring (Lines 275-278).
2-Although there is high standard deviation, all of the values are lower in 1D compared to 1A. Any explanation on this? why is this small drop?.
We have reviewed all the data bases after your comment. We have detected a serious error when transferring the values to apply the statistical processing for the first match. The values before the first game (1A) and after the same game (1D) were exchanged. We apologise for this error and thank the editor for highlighting this as we have been able to correct it in the paper (tables 2, 3 and 4)
3-The design of the study is valid and having 6 time points is acceptable. However, one wonders, if it is possible to include a new sample population and following the subjects for longer? Second paragraph of the ‘Discussion’ includes some information related to that. However, additional data can really support this manuscript by following the subjects for longer period of time. Do the authors consider studying another population of rugby players and following for longer hours after the match?.
We agree with your comment and consider it to be highly important to monitor the evolution of enzyme levels over a longer period. In this sense, we are already working on this with another population of rugby seven players.
4-Apart from these, the authors made several conclusions on the increase of several measurements after the 3rd match. For example, the increase in AST (41 to 71 in men, 13 to 21 in women). I am not convinced that these increases are physiologically meaningful since these values are still lover that the initial measurements in 1A (71 is less than 81 in men, 21 is less than 29 in women). Any explanation on these observations?.
The editor's concern has been addressed by correcting the error in the quantification of the samples from the first match. Once again we appreciate your comment.
5-Due to the above points listed, I invite the authors to tune-down some of the conclusion.
The conclusions have been rewritten.
6-Although the discussion session has a lot of useful information, I expected to find comparison of these values with healthy subjects. For, example what are the normal CK, LDH and AST values in the saliva of healthy individuals? Are there studies reporting these? Or do authors consider performing these analysis? Additionally, are there studies looking at these parameters in patients with muscle diseases? If so, what are these values. I think, both of these can show if the observed differences are biologically meaningful or not.
Unfortunately, there are no reference values for these markers in saliva. However, our group has published values of these 3 enzymes in saliva, comparing the levels in healthy people according to sample maintenance conditions (freezing temperature) and according to the subsequent process (Cenfrifuge speed and time) (Barranco T, Cerón JJ, López-Jornet P, Pastor J, Carrillo JM, Rubio M, Tornel PL, Cugat R, Tecles F, Tvarijonaviciute A. Impact of Saliva Collection and Processing Methods on Aspartate Aminotransferase, Creatin Kinase and Lactate Dehydrogenase Activities. Anal Sci. 2018;34(5):619-622. doi: 10.2116/analsci.17N035. PMID: 29743436.). When comparing the values of the means that were obtained in this study with values in our previous study, we observed that there are higher levels in the production of CK and AST enzymes in saliva throughout the competition. This comparison is described in the discussion section (Lines 200-207) for which the median values have been calculated and which have been added to the results (tables 2, 3 and 4).
7-A small point on the formatting of the study scheme. Please, make sure that all is in-line and readable (e.g. Match1, Match2, Match3).And the figure legend is missing.
In line with the reviewer’s indications, changes have been made throughout the whole article. A legend has been added to figure 1.
Reviewer 2 Report
Muscle damage could happen but it is not the norm.
Why the paper specifically refers to Rugby seven?
Contact cruelty in narrow spaces is usually in rugby 15. Rubgy 7 is more for speed fatigues instead of contact (compared to rugby 15).
Any difference depending on the role?
Maybe having analyzed rubgy 7 instead rubgy 15 makes the player more uniformly affected by the same contact cruelty and fatigue...
This could be a suggestion to justify your analysis referred to rugby 7 but this should be properly stated, explained, and proved.
Anyway, I believe that a proper differentiation of the data should be provided depend on the specific role of the player. The fatigue of the
scrum is obviously higher than that of three-quarter back or wing three-quarter
This further differentiation should be provided.
A personal curiosity: quite strange that the tests have been performed in May... no covid restriction in your league?
Overall the results have a minimum interest.
I invite the authors to collect more data and finding a more remarkable correlation among them.
Author Response
Reviewer 2
Muscle damage could happen but it is not the norm. Why the paper specifically refers to Rugby seven? Contact cruelty in narrow spaces is usually in rugby 15. Rubgy 7 is more for speed fatigues instead of contact (compared to rugby 15).
The clarification proposed by the reviewer was now added to the introduction (Lines 47-51).
Any difference depending on the role? Maybe having analyzed rubgy 7 instead rubgy 15 makes the player more uniformly affected by the same contact cruelty and fatigue... This could be a suggestion to justify your analysis referred to rugby 7 but this should be properly stated, explained, and proved.
We completely agree with the editor. We have introduced this information in the last paragraph of the introduction (Lines 77-78) justifying our interest in the analysis of rugby seven players.
Anyway, I believe that a proper differentiation of the data should be provided depend on the specific role of the player. The fatigue of the scrum is obviously higher than that of three-quarter back or wing three-quarter
This further differentiation should be provided.
Thank you for the comment, We indicate in section 2.3 Procedure (Material and methods), in particular lines 130 to 133, that the players’ position was not taken into account in this study due to the fact that physiological and performance characteristics in rugby seven are relatively homogeneous in the different positions of the game, based on results of the works of Higham D.G. et al., 2012 and Carreras D. et al., 2013. However, we have completed the information, justifying how the movement patterns in rugby seven are less position-dependant than in rugby 15. In this sense, as indicated in the Procedure section (Lines 134-135), we also verify that the match time was similar for all players, therefore no player played less than 7 minutes or more than 9 minutes.
A personal curiosity: quite strange that the tests have been performed in May... no covid restriction in your league?.
Apologies for not indicating the year. The competition took place in May 2019 (Line 103). We have added this information to the article.
Overall the results have a minimum interest. I invite the authors to collect more data and finding a more remarkable correlation among them.
Thank you for the suggestion. In this sense, we are currently developing new analyses with other rugby seven players to delve into the analysis of the obtained results by correlating them and other related enzymes with muscle damage. Our aim is to know more about the process of muscle fatigue and recovery.
Round 2
Reviewer 1 Report
I thank the authors for improving the manuscript.
Reviewer 2 Report
Paper has been improved